# IPG-Rec: Instance-aware Progressive Geometry Rectification for High-fidelity Sparse Novel View Synthesis

## Abstract

Sparse novel view synthesis is a challenging problem due to the limited information available. While recent 3D Gaussian Splatting (3DGS) approaches have leveraged monocular depth or diffusion priors to improve reconstruction quality, they struggle to generate multi-view consistent geometry efficiently. To address this problem, we propose an Instance-aware Progressive Geometry Rectification method, namely PG-Rec, to reconstruct the high-fidelity geometry from sparse inputs. Notably, our approach progressively and jointly optimizes 3D Gaussian representations by leveraging reliable pseudo-view images, along with instance-level and scene-level depth regularization, which promotes the reconstruction of high-fidelity 3D geometry with implicit cross-view semantic consistency. Considering insufficient information from sparse views, we employ instance-level and scene-level depth regularization to refine the 3D geometry cooperatively. The instance depth guides the 3D Gaussians to move toward their corresponding object, while the global depth maintains the relative spatial positions of Gaussians in different instances. With geometry refined by depth regularization, 3DGS renders more realistic images that guide diffusion to generate reliable pseudo-views. These pseudo-views are then used to further refine geometry. By combining depth regularization with high-fidelity pseudo-view rendering, our method progressively mitigates reconstruction defects from sparse inputs and acquires high-fidelity rendering images. Extensive experiments demonstrate that our PIGR outperforms current state-of-the-art methods in sparse novel view synthesis.

## 1 Introduction

Novel view synthesis (NVS) with sparse inputs aims to reconstruct photorealistic scenes, which presents a significant challenge in computer vision due to the limited information available. Methods such as Neural Radiance Field (NeRF) (Mildenhall et al., 2021) and 3D Gaussian Splatting (3DGS) (Kerbl et al., 2023a) have excelled in rendering high-quality novel views using dense and largely overlapping training images. However, in real-world scenarios where dense view information cannot be easily obtained, these methods risk producing unreliable artifacts due to the scarcity of scene information.

To prevent artifacts from appearing in novel views, regularization is crucial during geometry reconstruction. It helps to constrain the positions and colors of Gaussian primitives. Recently, depth constraints have gained significant attention as an effective method for improving reconstruction quality. For instance, DNGaussian (Li et al., 2024a) introduces patch-level depth

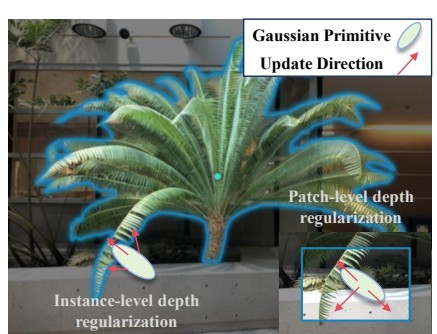

Figure 1: **Comparison of depth regularization methods.** Patch-level depth regularization often leads to ambiguities in positioning Gaussian primitives. In contrast, instance-level depth supervision enforces object-specific alignment, effectively promoting cross-view semantic consistency.

constraints that limit the movement of 3D Gaussian primitives. However, this approach divides the image into fixed-size patches, which can disrupt both semantic and geometric consistency—such as splitting a single object across multiple patches—and introduce object-level ambiguities (e.g., segmenting multiple objects within the same patch). These limitations hinder 3D Gaussian ellipsoids from accurately adapting to the correct objects, resulting in insufficient rendering fidelity and mutual interference between objects, yielding blurred and unreliable content in novel views.

Given the continuity of object distributions in 3D scenes, it is essential to impose geometry regularization at the instance level. To achieve this, we utilize object-level depth supervision from Depth Anything V2 (DA-V2) (Yang et al., 2024), a powerful large-scale monocular depth estimator. Using derived instance masks from SAM (Kirillov et al., 2023), the resulting depth maps demonstrate implicit cross-view semantic consistency. This signal guides 3D Gaussian primitives to align more accurately with the correct geometrical positions that cover each object. However, monocular depth estimation models typically predict relative depth values and rely solely on instance-level local depth regularization, which can lead to inconsistent depth scales across various objects. To rectify this issue, we incorporate global depth regularization to maintain a consistent depth scale for the entire scene and prevent erroneous relative updates of Gaussian ellipsoids between different instances. Thus, we propose a combined approach that integrates instance-level depth regularization with scene-level global depth supervision to improve the geometry of the 3D scene. The local depth guides the 3D Gaussian ellipsoids of each object to their correct positions in 3D space, while the global depth ensures that the relative spatial relationships among different instances are preserved.

Additionally, the sparsity of viewpoints limits the scene information, making it challenging to learn accurate 3D representations. Our aim is to optimize scene geometry with depth supervision while generating reliable pseudo-views from the current structure. These pseudo-views provide 2D image priors from unseen viewpoints, enhancing 3D representations. By using robust depth estimators, we extract reliable depth maps. Combining RGB and depth maps from pseudo-views improves scene geometry recovery. Inspired by diffusion-based generative models, we found that synthesized novel views from reliable viewpoints, although slightly imperfect, still offer sufficient depth cues to optimize 3D geometry. We use both RGB and depth maps to correct scene geometry, with depth regularization enhancing accuracy while realistic renderings provide useful pseudo-supervision. This combined approach helps repair reconstruction flaws caused by sparse viewpoints, significantly improving geometric fidelity and enabling realistic novel view synthesis. The proposed IPG-Rec, combined with an instance-aware progressive geometry rectification framework, can generate high-fidelity novel views while addressing geometric defects and viewpoint sparsity, achieving state-of-the-art performance.

Our primary contributions are as follows:

- We propose IPG-Rec, an Instance-aware Progressive Geometry Rectification framework that jointly optimizes 3D Gaussian representations by integrating reliable pseudo-view images with multi-level depth regularization. This collaborative optimization enables the reconstruction of high-fidelity geometry constrained by implicit cross-view semantic consistency from sparse inputs.
- We introduce a dual-depth regularization strategy that combines instance-level supervision to guide Gaussians toward their corresponding objects and scene-level supervision to maintain relative spatial consistency across objects. This design enforces both local precision and global structural coherence, effectively mitigating reconstruction defects caused by sparse views.
- Extensive experiments demonstrate the superiority of IPG-Rec against existing methods for sparse novel view synthesis.

## 2 RELATED WORK

### 2.1 FEW-SHOT NOVEL VIEW SYNTHESIS

Novel view synthesis (NVS) aims to generate images from unobserved viewpoints based on a limited set of input views. Neural Radiance Fields (NeRF) achieve high-quality rendering through volumetric techniques, but they typically require numerous input views, making it challenging to balance

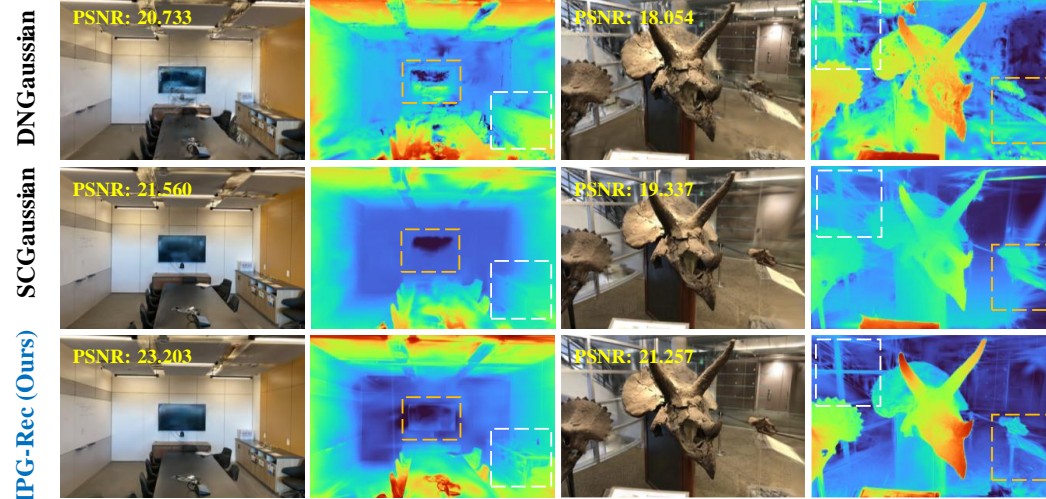

Figure 2: **Comparison of novel view synthesis and corresponding geometry rendering with 3 training views.** The proposed method produces novel views with more consistent geometry and improved realism.

quality and efficiency. Recently, a few practical, few-shot methods have emerged to handle sparse inputs, which can be grouped into two main approaches. Firstly, applying NeRF with regularization strategies (Niemeyer et al., 2022b); (Wang et al., 2023a), incorporating depth supervision (Roessle et al., 2022), geometric and appearance constraints (Wang et al., 2023a), and knowledge distillation (Kim, 2023), enables scene reconstruction from sparse viewpoints. While these methods benefit from NeRF's strong representation ability, decreasing the number of input views can still degrade synthesis quality, and the computational cost of the backbone limits efficiency. The second approach relies on 3D Gaussian Splatting (3DGS) or generative methods. Initially proposed by Kerbl et al. (Kerbl et al., 2023a), 3DGS explicitly models scenes with Gaussian primitives, enhancing training and inference speed. Subsequent works (Li et al., 2024a); (Xu et al., 2024) add additional constraints to improve sparse-view reconstruction, such as monocular supervision (FSGS (Zhu et al., 2023)), floating-primitive pruning (SparseGS (Xiong et al., 2023)), primitive-view and motion constraints (CoherentGS (Paliwal et al., 2024)), and depth regularization (DNGaussian (Li et al., 2024a)). 3DGS-based methods can achieve near-NeRF reconstruction quality with fewer views while reducing computational costs, but it remains challenging to obtain globally consistent reconstruction with sparse inputs.

## 2.2 DEPTH REGULARIZATION FOR NVS

To improve the generalization of Neural Radiance Fields (NeRF) with sparse viewpoints while balancing reconstruction quality and inference speed, various prior-based strategies have been proposed. These include semantic similarity (Fridovich-Keil et al. (2022)), normal vectors (Wang et al. (2022)), and ground-truth or estimated depth information. Among these, depth-prior-based approaches are particularly effective in enhancing reconstruction quality. DSNeRF (Deng et al., 2022); GeoNeRF (Johari et al., 2022); ENeRF (Wang et al., 2023b), incorporate ground-truth depth to impose geometric constraints, while SCADE (Uy et al., 2023) and FSGS (Zhu et al., 2023) introduce a 3DGS densification strategy based on a zero-shot depth estimator, and NerfingMVS (Wei et al., 2021) employs depth reconstructed via the motion recovery framework COLMAP to train depth predictors. Monocular depth-based methods, however, face limitations when input views are sparse and can be unstable. To address these issues, SparseNeRF (Wang et al., 2023a); NeRDi (Deng et al., 2023), and NeuralLift-360 (Xu et al., 2023) leverage pre-trained depth models to obtain robust depth ordering information, thereby enhancing spatial coherence, while RegNeRF (Niemeyer et al., 2022b) and DONeRF (Neff et al., 2021) incorporate geometric regularization, intelligent sampling, and probabilistic depth supervision, respectively, to improve rendering quality. Despite their effectiveness, depth priors used for regularization are often too coarse, and inpainting may introduce artifacts, both of which can reduce realism.

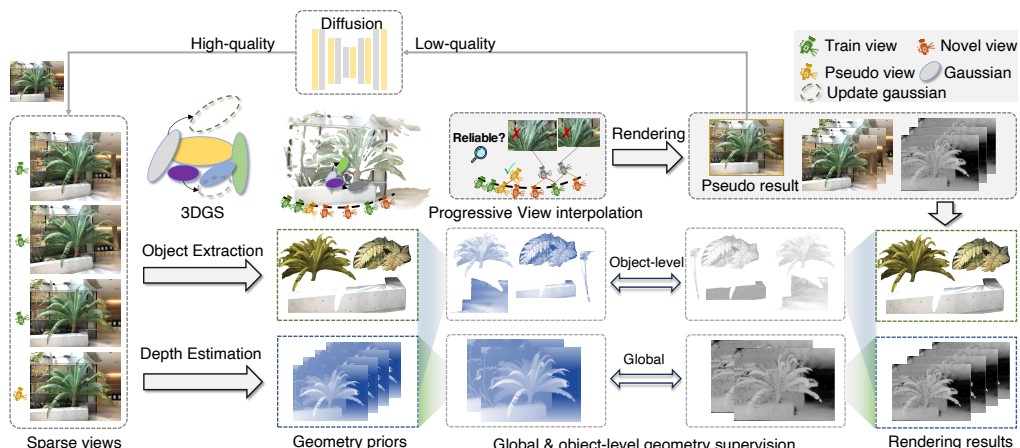

Figure 3: IPG-Rec pipeline. Our method follows the procedure outlined below. First, we train the 3DGS model in the early training stage with RGB image supervision, along with the proposed instance-level depth and global regularization. Next, we employ a one-step pre-trained diffusion model to eliminate artifacts of pseudo-view generated progressively over a few iterations. The resulting clean images are then incorporated into the training set and used for both image and depth supervision to rectify fine-grained and consistent geometry of a scene.

## 2.3 DIFFUSION PRIORS FOR NVS

Generative models, leveraging their strong prior knowledge, are increasingly applied to improve novel view synthesis. Some studies employ diffusion models to generate neural radiance fields (NeRFs) for optimizing rendering results. For instance, DreamFusion (Poole et al., 2022) enhances view synthesis through language guidance, GAUDI (Bautista et al., 2022) trains diffusion models in the NeRF latent space, and NeRDI (Deng et al., 2023) constrains novel view distributions using features from input images. Other approaches achieve zero-shot view synthesis conditioned on images and poses via model fine-tuning (Chan et al., 2023); (Tung et al., 2024); (Yu et al., 2024), or perform inpainting on warped images using text-to-image models (Li et al., 2024b); (Engstler et al., 2025). However, these methods often suffer from artifacts and accumulated errors in general scenes. In addition, methods such as Deceptive-NeRF (Liu et al., 2023);3DGS-Enhancer (Liu et al., 2024); and DIFIX3D+ (Wu et al., 2025) leverage diffusion priors to enhance "pseudo-observations" rendered from 3D representations, expanding the training set and fine-tuning the 3D representations. Nevertheless, they still face challenges in removing artifacts and maintaining long-range consistency.

## 3 METHOD

In this section, we present the details of the proposed IPG-Rec algorithm, as summarized in Algorithm 1. The whole pipeline of our model is illustrated in Figure 3. The IPG-Rec incorporates 2D image priors progressively and applies depth regularization continuously, enabling joint optimization of 3D geometry. In Sec. 3.1, we first review 3D Gaussian Splatting (3DGS). Then, we introduce depth-induced geometry rectification in Sec. 3.2, which includes instance-level depth regularization and scene-level depth regularization, and then we introduce diffusion-based progressive pseudo-view Geometry refinement in Sec. 3.3. The full loss and training details in Sec . 3.4.

The IPG-Rec framework jointly optimizes 3D Gaussian representations by integrating reliable pseudo-view images with multi-level depth regularization. This collaborative optimization enables the reconstruction of high-fidelity geometry constrained by implicit cross-view semantic consistency from sparse inputs.

### 3.1 PRELIMINARY OF 3D GAUSSIAN SPLATTING

3D Gaussian Splatting (Kerbl et al., 2023b) explicitly models a scene through a collection of anisotropic 3D Gaussian primitives, each defined by a center $\mu \in \mathbb{R}^3$, an anisotropic covariance matrix $\Sigma \in \mathbb{R}^{3\times3}$, an alpha value $\alpha \in [0,1]$ representing opacity, and spherical harmonics coefficients (SH). Given a 3D position $x \in \mathbb{R}^3$, the probability density function of 3D Gaussian is defined as equation (1), in which $(\cdot)^T$ represents a transpose operation and $(\cdot)^{-1}$ denotes matrix inversion.

$$G(x) = e^{-\frac{1}{2}(x-\mu)^T\Sigma^{-1}(x-\mu)} \tag{1}$$

To render 3D Gaussians in 2D, we project their mean positions by point projection, and project their covariance using the following equation (2), where $\mathbf{W} \in \mathbb{R}^{3\times3}$ is the viewing transformation and $\mathbf{J} \in \mathbb{R}^{3\times3}$ is the Jacobian of the affine approximation of the projective transformation.

$$\mathbf{\Sigma}' = \mathbf{JW}\,\mathbf{\Sigma}\,\mathbf{W}^T\mathbf{J}^T \tag{2}$$

To optimize covariance matrices, we use an equivalent representation (3), in which $\mathbf{R} \in \mathbb{R}^{3\times3}$ and $\mathbf{S} \in \mathbb{R}^{3\times3}$ are rotation and scaling matrices, respectively.

$$\Sigma = \mathbf{RSS}^T\mathbf{R}^T \tag{3}$$

Gaussian Splatting also includes spherical harmonics coefficients to model the appearance of the scene. Gradients for all parameters are derived explicitly to avoid overhead during training. Each Gaussian encodes the color $c$ using spherical harmonics, which gives a value depending on the viewing directions. The $\alpha$−blending point-based rendering for a pixel color $\mathbf{c}$ is done by blending $\mathcal{N}$ points in the depth order from front to back. The definition of $\mathbf{c}$ is given in Equation (4), where $\alpha_i$ is given by a 2D Gaussian multiplied by a learned per-Gaussian opacity. Besides, to achieve better geometry, we compute camera parameters and 3D point clouds using an off-the-shelf structure-from-motion system, COLMAP (Schönberger & Frahm, 2016; Schönberger et al., 2016).

$$\mathbf{c} = \sum_{i\in\mathcal{N}} \mathbf{c}_i\alpha_i \prod_{j=1}^{i-1}(1-\alpha_j), \tag{4}$$

### 3.2 DEPTH-INDUCED GEOMETRY RECTIFICATION

**Instance-level Depth Regularization.** Incorporating a few 2D pseudo-view images is not sufficient for 3D optimization. To learn the fine-grained geometry, existing local depth regularization for 3DGS employs fixed-size patches. However, these patches often span across object boundaries and include multiple instances simultaneously, leading to the depth variance within a patch being typically large. Moreover, the depth values of the background near object edges are typically set to zero during normalization, which hinders the accurate reconstruction of irregular object boundaries. In addition, this approach does not focus on semantic cues within the scene and can hardly reconstruct well for promoting cross-view geometric consistency. To mitigate this issue, we introduce an object-level depth regularization method that computes the difference of each segmented irregular object region in both the monocular and predicted depth maps by the 3DGS model. This instance-aware approach enables 3D Gaussians to rectify the fine-grained geometry in local regions and maintain semantic consistency across views. Specifically, given an RGB image $\mathcal{I} \in \mathbb{R}^{3\times H\times W}$, we employ the pre-trained SAM (Kirillov et al., 2023) to segment the picture and obtain a set of instance-level masks. Then, we binarize each mask by initializing a tensor of size $H \times W$ filled with zeros. The pixel positions corresponding to a mask are set to 1, while all other positions remain 0. In this way, we obtain $n$ mask maps, denoted as $M = \{M_1, M_2, ..., M_n\}$, where the pixels with a value of 1 in each mask represent the same instance, such as a flower, a leaf, or a house.

Moreover, we adopt the Depth Anything V2 (Yang et al., 2024) pretrained on a large-scale dataset to predict the monocular depth $D_{gt} \in \mathbb{R}^{H\times W}$ for each input image, enabling finer and more robust depth regularization. Since the estimated monocular depth maps are relative values, whereas the rendered depths are COLMAP-anchored. Inspired by (Xiong et al., 2023), we incorporate Pearson correlation across the same instance region between monocular and rendered depth maps to compute similarity. Utilizing the Pearson correlation loss promotes high cross-correlation between the same objects in both depth maps, regardless of variations in depth value ranges. Thus, the instance-aware depth loss between the monocular depth map and the rendered depth map $D_{re} \in \mathbb{R}^{H\times W}$

is computed after applying each object mask $M_i$ to the corresponding regions in both depth maps. This process is illustrated in Equations (5), where $D'_{gt}, D'_{re} \in \mathbb{R}^{H \times W}$ denote the ground-truth and rendered depth values within the same object region, respectively. Depth values outside this region are set to zero and excluded from the loss computation. The $L_{obj}$ represents the instance-aware loss, and the definition of $PCC(X, Y)$ follows (Xiong et al., 2023).

$$D'_{gt} = M_i \cdot D_{gt},$$
$$D'_{re} = M_i \cdot D_{re},$$
$$L_{obj} = \frac{1}{n} \sum_{i \in n} (1 - PCC(D'_{gt}, D'_{re})),$$

$$PCC(X, Y) = \frac{E[XY] - E[X]E[Y]}{\sqrt{E[Y^2] - E[Y]^2} \sqrt{E[X^2] - E[X]^2}}.$$

(5)

**Scene-level Depth Regularization.** We observed that segmentation models may struggle to accurately segment the corresponding masks due to the presence of numerous small background objects and occlusions. Moreover, during the early stages of training, the limited 3D representation capability of 3DGS leads to poor-quality rendered depth maps. Thus, calculating local depth loss within each object mask is insufficient to overall geometry rectification. To address these issues and improve the rendering ability of 3D Gaussian primitives in the scene, we introduce global depth supervision for the entire image. Similar to the instance-aware loss, the global depth loss is formally defined as equation (6), where $D_{gt}$ is the monocular depth and $D_{re}$ is the rendered depth by 3D Gaussians. Intuitively, the $L_{global}$ facilitates the optimization of holistic scene geometry, thereby improving the fidelity of the 3D representation.

$$L_{global} = \frac{1}{n} \sum_{i \in n} (1 - PCC(D_{gt}, D_{re})). \quad (6)$$

---

**Algorithm 1** Progressive Instance-aware Geometry Rectification

---

**Input:** sparse views $\{I_i\}_{i=1}^{N_{sparse}}$, 3DGS model $\mathcal{R}$, Monocular depth map $\mathcal{D}_{gt}$, Target views $\{\mathcal{V}_i\}_{i=1}^{N_{sparse}}$, Reference views $\mathcal{V}_r$, Pretrianed one-step diffusion model $D$, Perturbation step size $\Delta_{pose}$, Refinement steps $\{s_1, s_2, ..., s_n\}$
**Output:** High-fidelity Novel views $\mathcal{V}_{novel}$.

1: # *Optimize 3D gaussian primitives*
2: Compute mask maps $\{M_i\}_{i=1}^{n}$ of the input image
3: Initialize pseudo pose set $\mathcal{P} = \emptyset$
4: **while** $step < iterations$ **do**
5:      *# generate pseduo camera poses*
6:      **if** step $== s_1$ **then**
7:          **for** $j = 1$ to $N_{sparse}$ **do**
8:              pseudo pose $p_j \leftarrow \mathcal{I}_j + \Delta_{pose}$
9:              $\mathcal{P}_{pose} = p_j \cup \mathcal{P}_{pose}$
10:      **if** step $\in \{s_2, s_3, ..., s_n\}$ **then**
11:          **for** $j = 1$ to $N_{sparse}$ **do**
12:              pick $\mathcal{P}_{near, j} \in \mathcal{P}_{pose}$ nearest to $\mathcal{V}_i$
13:              pseudo pose $p_j \leftarrow \mathcal{P}_{near, j} + \Delta_{pose}$
14:          **for** each pseudo pose **do**
15:              Render pseudo-view images $\mathcal{I}_{pse}$
16:              $\mathcal{I}_{clean} \leftarrow \mathcal{D}(\mathcal{I}_{pse})$
17:              $\{I_i\}_{i=1}^{N+1} \leftarrow \{I_i\}_{i=1}^{N} \cup \mathcal{I}_{clean}$
18:      Compute $L_{rgb}, L_{obj}, L_{global}, L$ with $\mathcal{D}_{gt}$
19:      Update $\mathcal{R}$ via gradient descent

---

### 3.3 DIFFUSION-BASED PROGRESSIVE PSEUDO-VIEW GEOMETRY REFINEMENT

We employ the generative image priors of SD-Turbo (Sauer et al., 2024), a pre-trained one-step diffusion model, which enables high efficiency of our method. Following DIFIX3D+ (Wu et al., 2025), we find that the difix model, which was pre-trained on the DL3DV (Ling et al., 2024) benchmark dataset, exhibits impressive performance on eliminating Gaussian artifacts from our generated pseudo views, as shown in Figure 5. Thus, we utilize the pre-trained difix model as a fixer to eliminate the artifacts of pseudo-view images and obtain clean pseudo-images.

Specifically, we first optimize the 3DGS model in the early training stage to learn the basic 3D structure, and then add pseudo-views in fixed steps. From each scene dataset, we compute the $N_{sparse}$ validation views that exhibit the greatest discrepancy from the $N_{sparse}$ sparse inputs, ensuring that the progressively generated pseudo-views contain more under-observed image priors information. Furthermore, to constrain the difix model to generate content pertinent only to the current scene, we select a reference image with a viewpoint similar to that of each newly generated viewpoint. Both the reference and the Gaussian-rendered pseudo-image are input into the difix model for denoising, producing clean pseudo-images, which are then added to the training set for 2D image augmentation.

Figure 4: **Qualitative comparisons on LLFF (the left three columns) and IBRNet (the right two columns) datasets with three training views. The reconstruction of our method is more accurate and exhibits finer details.**

### 3.4 OVERALL PIPELINE

**Loss function.** The loss function consists of three parts (equation (7)): the original photometric loss $L_{rgb}$(equation (8)), the global depth regularization loss $L_{global}$(equation (6)), and the instance-level depth regularization loss $L_{obj}$ (equation (5)). During training, we set $\beta = 0.1$, $\delta = 0.1$, and $\lambda = 0.2$.

$$L = L_{rgb} + \beta L_{global} + \delta L_{obj}, \tag{7}$$

$$L_{rgb} = L_1(\hat{I}, I) + \lambda L_{ssim}(\hat{I}, I). \tag{8}$$

## 4 EXPERIMENTS

### 4.1 SETTINGS.

Table 1: **Quantitative comparisons on the LLFF and IBRNet datasets with three training views.** Best results are in **bold**. We run our method 5 times and report the error bar in the appendix.

| Method | Approach | LLFF | | | | IBRNet | | | |
|---|---|---|---|---|---|---|---|---|---|
| | | PSNR ↑ | SSIM ↑ | LPIPS ↓ | AVG ↓ | PSNR ↑ | SSIM ↑ | LPIPS ↓ | AVG ↓ |
| Mip-NeRF (Barron et al., 2021) | NeRF-based | 14.62 | 0.351 | 0.495 | 0.246 | 15.83 | 0.406 | 0.488 | 0.223 |
| RegNeRF (Niemeyer et al., 2022a) | | 19.08 | 0.587 | 0.336 | 0.149 | 19.05 | 0.542 | 0.377 | 0.152 |
| FreeNeRF (Yang et al., 2023) | | 19.63 | 0.612 | 0.308 | 0.134 | 19.76 | 0.588 | 0.333 | 0.135 |
| SparseNeRF (Wang et al., 2023a) | | 19.86 | 0.624 | 0.328 | 0.127 | 19.90 | 0.593 | 0.364 | 0.137 |
| 3DGS (Kerbl et al., 2023a) | 3DGS-based | 16.46 | 0.440 | 0.401 | 0.192 | 17.79 | 0.538 | 0.377 | 0.166 |
| FSGS (Zhu et al., 2023) | | 20.43 | 0.682 | 0.248 | - | 19.84 | 0.648 | 0.306 | 0.130 |
| DNGaussian (Li et al., 2024a) | | 19.12 | 0.591 | 0.294 | 0.132 | 19.01 | 0.616 | 0.374 | 0.151 |
| SCGaussian (Peng et al., 2024) | | 20.41 | 0.705 | 0.218 | 0.105 | 21.59 | **0.731** | 0.233 | 0.097 |
| DropGaussian (Park et al., 2025) | | 20.76 | 0.713 | 0.200 | - | - | - | - | - |
| **IPG-Rec (Ours)** | | **21.44** | **0.731** | **0.183** | **0.078** | **22.14** | 0.700 | **0.205** | **0.091** |

**Datasets and Metrics.** We conduct our experiments on three datasets representing generic scenes with complex textures: LLFF (Mildenhall et al., 2019), Tanks and Temples (T&T) (Knapitsch et al., 2017), and IBRNet (Wang et al., 2021). The LLFF dataset is a forward-facing dataset that contains eight complex real-world scenes and multiple irregular objects, where camera trajectories exhibit limited variation. For LLFF and IBRNet dataset partitioning, we adopt a uniform sampling strategy. Specifically, every 8th image is selected and assigned to the validation set. From the remaining images, we again uniformly sample every 8th frame for the training set, while all remaining images are allocated to the test set. Following (Peng et al., 2024; Li et al., 2024a), we use 9 scenes for

evaluation. T&T is a large-scale dataset collected from challenging real-world environments, encompassing both indoor and outdoor scenes. We use 8 scenes for evaluation and follow the same splitting protocol as LLFF. We report PSNR, SSIM, and LPIPS scores to evaluate our reconstruction performance following previous works (Li et al., 2024a; Peng et al., 2024). Additionally, we also report the geometric average (AVG) of $\mathrm{MSE} = 10^{-\mathrm{PSNR}/10}$, $\sqrt{1 - \mathrm{SSIM}}$, and LPIPS as in (Niemeyer et al., 2022a). The detailed settings of these three datasets are provided in A.1 of the Appendix.

**Baselines.** Following the prior works (Li et al., 2024a; Peng et al., 2024), we compare our model against both NeRF-based and 3DGS-based few-shot NVS methods. We take current classic approaches like Mip-NeRF (Barron et al., 2021), RegNeRF (Niemeyer et al., 2022a), RegNeRF (Niemeyer et al., 2022a), SparseNeRF (Wang et al., 2023a), 3DGS (Kerbl et al., 2023a), FSGS (Zhu et al., 2023), DNGaussian (Li et al., 2024a), SCGaussian (Peng et al., 2024) and DropGaussian (Park et al., 2025) for comparison. We reproduce the source code of SCGaussian in LLFF and T&T datasets.

**Implementation Details.** We utilize the pre-trained SAM ViT-B/16 model to segment the masks of each image for instance-level depth regularization. For each scene, we train it for 10,000 iterations to optimize the parameters of 3D Gaussians. We set the perturbation steps to 3000, 5000, 7000 to generate pseudo views for both image and depth augmentation. All experiments are implemented on an NVIDIA A800 GPU.

Table 2: **Quantitative comparisons on the T&T dataset with 3 training views.**

| Method | Approach | PSNR ↑ | SSIM ↑ | LPIPS ↓ |
|---|---|---|---|---|
| MipNeRF (Barron et al., 2021) | | 12.57 | 0.241 | 0.623 |
| RegNeRF (Niemeyer et al., 2022a) | NeRF-based | 13.12 | 0.268 | 0.618 |
| FreeNeRF (Yang et al., 2023) | | 12.30 | 0.308 | 0.636 |
| SparseNeRF (Wang et al., 2023a) | | 13.66 | 0.331 | 0.615 |
| 3DGS (Kerbl et al., 2023a) | | 17.14 | 0.493 | 0.397 |
| FSGS (Zhu et al., 2023) | | 20.01 | 0.652 | 0.323 |
| DNGaussian (Li et al., 2024a) | 3DGS-based | 18.59 | 0.573 | 0.437 |
| SCGaussian (Peng et al., 2024) | | 20.98 | 0.703 | 0.303 |
| **IPG-Rec (Ours)** | | **21.38** | **0.707** | **0.286** |

## 4.2 PERFORMANCE EVALUATION.

**Results on LLFF, IBRNet and T&T Datasets.** Following prior works (Park et al., 2025; Peng et al., 2024; Li et al., 2024a), we employ the aforementioned split method to sample three images for training. The quantitative results on three datasets with recent state-of-the-art (SOTA) methods are summarized in Tables 1 and 2. All results demonstrate that the proposed IPG-Rec achieves SOTA performance on most of the metrics across multiple real-world scenes. It is worth noting that in LLFF and T&T datasets, which contain numerous small background objects and irregular scene structures with substantial semantic variations among internal objects of scenes, our proposed method achieves significant improvements across all metrics. As illustrated in Figure 2 and 4, the reconstruction comparisons with top two SOTA approaches show that our approach accurately reconstruct fine and sharp object boundaries embedded in complex backgrounds, such as the railings of distant spiral staircases, the edges of tiny blades of grass, and the intricate structures of flower stamens. The qualitative results of our method consistently surpass recent baselines and nearly reproduce the ground-truth viewpoints. Together, these strong quantitative and qualitative results demonstrate that our proposed instance-level depth regularization effectively guides 3D Gaussians to learn fine-grained local geometry details. Moreover, the integration of unseen 2D image priors with monocular-depth-based supervision provides complementary optimization, enabling 3D Gaussian representations to capture high-fidelity geometry with latent cross-view semantic consistency. In addition, we provide the comparisons between our IPG-Rec and the top two approaches Peng et al. (2024); Li et al. (2024a) on each subset of the LLFF dataset in A.2 of the Appendix.

## 4.3 ABLATION STUDY

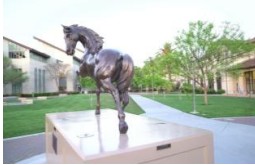 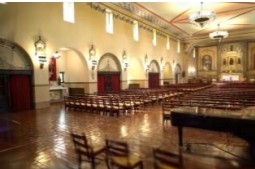 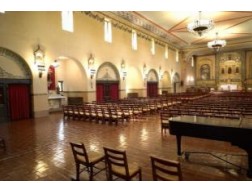

| **Pseudo View** | **Clean View** | **Pseudo View** | **Clean View** |

Figure 5: **The visualization of eliminating artifacts with a pre-trained difix model.**

Table 3: **Ablation studies on LLFF Dataset. Bold font indicates the better results.**

| Method | PSNR ↑ | SSIM ↑ | LPIPS ↓ |
|---|---|---|---|
| Baseline | 18.804 | 0.595 | 0.305 |
| W/ scene-level depth | 18.883 | 0.585 | 0.304 |
| W/ instance-level depth | 19.152 | 0.622 | 0.271 |
| W/ progressive pesudo views | **21.442** | **0.731** | **0.183** |

Table 4: **Using different depth regularization for sparse novel pose synthesis on LLFF dataset.**

| Method | PSNR ↑ | SSIM ↑ | LPIPS ↓ |
|---|---|---|---|
| Our Backbone | 20.406 | 0.712 | 0.213 |
| Gaussian Rendered depth | 20.145 | 0.646 | 0.297 |
| Monocular depth (Depth Anything V2) | **21.442** | **0.731** | **0.183** |

To evaluate the effectiveness of each individual strategy in the IPG-Rec algorithm, we conducted ablation studies on the LLFF dataset, with the results summarized in Table 3. Specifically, we ablate the scene-level depth regularization, the instance-level local depth regularization, and the progressive 2D pseudo views augmentation. On the one hand, integrating the baseline with the instance-level depth regularization strategy improved the PSNR by 0.079 and reduced the LPIPS by 0.001. On the other hand, adding progressive pseudo views on top of the baseline DNGaussian (Li et al., 2024a) increased PSNR and SSIM by 0.348 and 0.027,

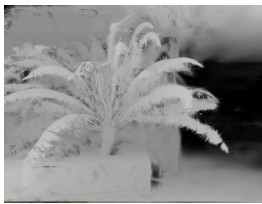 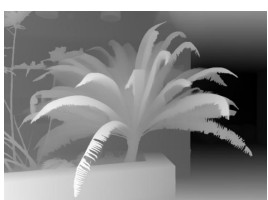

| **Rendered** | **Depth Anything V2** |

Figure 6: **Comparison of Depth Estimations.** The depth estimates from Depth Anything V2 (Yang et al., 2024) serve as a more effective optimization constraint due to their superior smoothness and accuracy over rendered depth.

respectively, while decreasing LPIPS by 0.034, demonstrating that our pseudo-label generation strategy significantly enhances novel view synthesis. Finally, combining scene-level and instance-level depth regularization with progressive pseudo views resulted in improvements of 2.638, 0.136, and 0.122 in PSNR, SSIM, and LPIPS, respectively, indicating that these proposed strategies complement each other and enable IPG-Rec to achieve new state-of-the-art performance in novel view synthesis.

Besides, on the LLFF dataset, we compared our backbone with the depth map supervised by Gaussian rendering and directly with the depth extracted by the pre-trained, powerful depth estimation model Depth Anything V2. The depth map from the Gaussian rendering is relatively poor. It can be seen in Table 4 that Depth Anything V2 Yang et al. (2024) provides smoother and more accurate depth estimation compared to Gaussian rendered depth, making it a more suitable constraint for model optimization. The comparison between the two approaches is shown in Figure 6.

## 5 CONCLUSION

In this work, we propose IPG-Rec, a novel view synthesis algorithm for 3D scenes from sparse input views. The method incorporates local instance-aware and global multi-view depth regularization strategies, along with a one-step diffusion model that progressively supplements unseen view data. IPG-Rec achieves new state-of-the-art performance on the task of sparse-view based novel view synthesis, delivering superior results on popular benchmarks including LLFF Mildenhall et al. (2019), IBRNet Wang et al. (2021), and T&T Knapitsch et al. (2017), with PSNR scores of 21.44, 22.14, and 21.38, respectively. The approach enables high-fidelity, fine-grained rendering with multi-view consistency in complex scenes while maintaining efficiency and strong generalization capability. It effectively mitigates challenges associated with irregular object instances and intricate textures in real-world sparse-view 3D reconstruction.

## 6 REPRODUCIBILITY STATEMENT

We have made extensive efforts to ensure the reproducibility of our work. The implementation details of IPG-Rec, including network architecture, training strategies, and optimization settings, are provided in Section X of the main paper and further elaborated in the appendix. Complete descriptions of the datasets (LLFF, IBRNet, etc.), data preprocessing steps, and the evaluation protocol are also included in the supplementary materials. To facilitate reproduction of our results, we will release the anonymized source code, training scripts, and configuration files as part of the supplementary submission. In addition, we provide proofs of the theoretical claims in the appendix and ablation studies that validate the effectiveness of each component, including instance-level and scene-level depth regularization. Together, these resources should allow researchers to fully reproduce and extend our results in sparse novel view synthesis.

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

# A APPENDIX

You may include other additional sections here.

## A.1 BENCHMARK DATASETS

We evaluate our algorithm on three benchmark datasets: LLFF (Mildenhall et al., 2019), IBR-Net (Wang et al., 2021), and Tanks and Temples (T&T) (Knapitsch et al., 2017). The LLFF dataset consists of scenes such as Fern, Flower, Horns, Room, Leaves, Orchids, Trex, and Fortress (Mildenhall et al., 2019). The IBRNet dataset includes scenes like Giraffe Plush, Yamaha Piano, Sony Camera, Japanese Camellia, Scaled Model, Dumbbell Jumprope, Hat on Fur, Roses, and Plush Toys (Wang et al., 2021). For both LLFF and IBRNet, every 8th image is held out for testing, and three sparse views are uniformly sampled from the remaining images for training. The T&T dataset contains a variety of indoor and outdoor complex scenes, including Ballroom, Barn, Church, Family, Francis, Horse, Ignatius, and Museum (Knapitsch et al., 2017). The same training and testing protocol is applied: every 8th image is reserved for testing, while three sparse views are uniformly sampled from the rest for training. Quantitative results in terms of PSNR, SSIM, and LPIPS are reported for all three datasets.

## A.2 MORE EXPERIMENTAL DETAILS

The comparison of IPG-Rec and DNGaussian (Li et al., 2024a) across LLFF subcategories is presented in Table 5. IPG-Rec demonstrates particularly strong performance on "Flower," "Fortress," and "Horns." For example, on the "Fortress" scene, PSNR and SSIM improved by 5.740 and 0.405, respectively. Across all subcategories, the average improvements in PSNR and SSIM were 2.638 and 0.136, respectively. Similarly, we report the performance of IPG-Rec and SCGaussian (Peng et al., 2024) on the LLFF dataset, as shown in Table 6. For the Flower, Fortress, Horns, Room, and Trex subsets, IPG-Rec achieves a PSNR improvement of more than 1.2 over SCGaussian. In terms of SSIM, IPG-Rec shows a notable advantage on Horns and Trex, with gains of approximately 0.6. Moreover, for these two subsets, our method reduces the LPIPS score by more than 0.03 compared to SCGaussian.The proposed IPG-Rec algorithm exhibits strong robustness and generalization capability across different subcategories.

Table 5: **Performance comparison of the proposed IPG-Rec and DNGaussian (Li et al., 2024a) on the LLFF subcategories. Blue font indicates substantial improvement of the proposed IPG-Rec.**

| Subcategory | DNGaussian | | | IPG-Rec (Ours) | | | | | |
| | PSNR ↑ | SSIM ↑ | LPIPS ↓ | PSNR ↑ | | SSIM ↑ | | LPIPS ↓ | |
| --- | --- | --- | --- | --- | --- | --- | --- | --- | --- |
| Fern | 20.670 | 0.672 | 0.268 | 22.506 | (+1.836) | 0.749 | (+0.077) | 0.232 | (-0.036) |
| Flower | 19.438 | 0.584 | 0.299 | 22.781 | (+3.343) | 0.721 | (+0.137) | 0.209 | (-0.090) |
| Fortress | 20.335 | 0.441 | 0.380 | 26.075 | (+5.740) | 0.846 | (+0.405) | 0.130 | (-0.250) |
| Horns | 18.054 | 0.616 | 0.353 | 21.257 | (+3.203) | 0.768 | (+0.152) | 0.206 | (-0.147) |
| Leaves | 16.427 | 0.548 | 0.279 | 17.159 | (+0.732) | 0.535 | (-0.013) | 0.348 | (+0.069) |
| Orchids | 14.630 | 0.405 | 0.347 | 17.210 | (+2.580) | 0.558 | (+0.153) | 0.276 | (-0.071) |
| Room | 20.733 | 0.779 | 0.270 | 23.203 | (+2.470) | 0.852 | (+0.073) | 0.184 | (-0.086) |
| Trex | 20.144 | 0.715 | 0.242 | 21.347 | (+1.203) | 0.817 | (+0.102) | 0.185 | (-0.057) |
| Average | 18.804 | 0.595 | 0.305 | 21.442 | (+2.638) | 0.731 | (+0.136) | 0.221 | (-0.084) |

## A.3 MORE DISCUSSION ON DEPTH REGULARIZATION

As shown in Figure 7, compared with DNGaussian (Li et al., 2024a), our proposed instance-level local depth regularization captures sharp edges of irregular objects in complex textures and fine background regions, enabling the recovery of high-fidelity object appearance and geometric structure.

Table 6: **Performance comparison of the proposed IPG-Rec and SCGaussian (Peng et al., 2024) on the LLFF subcategories. Blue font indicates substantial improvement of the proposed IPG-Rec over the SOTA method.**

| Subcategory | SCGaussian | | | IPG-Rec (Ours) | | | | | |
|---|---|---|---|---|---|---|---|---|---|
| | PSNR ↑ | SSIM ↑ | LPIPS ↓ | PSNR | ↑ | SSIM | ↑ | LPIPS | ↓ |
| Fern | 22.131 | 0.733 | 0.191 | 22.506 | (+0.375) | 0.749 | (+0.016) | 0.232 | (+0.041) |
| Flower | 21.512 | 0.688 | 0.223 | 22.781 | (+1.269) | 0.721 | (+0.033) | 0.209 | (-0.014) |
| Fortress | 24.480 | 0.810 | 0.139 | 26.075 | (+1.595) | 0.846 | (+0.036) | 0.130 | (-0.009) |
| Horns | 19.337 | 0.709 | 0.246 | 21.257 | (+1.920) | 0.768 | (+0.059) | 0.206 | (-0.040) |
| Leaves | 17.625 | 0.624 | 0.271 | 17.159 | (-0.466) | 0.535 | (-0.089) | 0.348 | (+0.077) |
| Orchids | 16.490 | 0.530 | 0.256 | 17.210 | (+0.720) | 0.558 | (+0.028) | 0.276 | (+0.020) |
| Room | 21.560 | 0.849 | 0.157 | 23.203 | (+1.643) | 0.852 | (+0.003) | 0.184 | (+0.027) |
| Trex | 20.114 | 0.754 | 0.218 | 21.347 | (+1.233) | 0.817 | (+0.063) | 0.185 | (-0.033) |
| Average | 20.406 | 0.712 | 0.213 | 21.442 | (+1.036) | 0.731 | (+0.019) | 0.221 | (+0.009) |

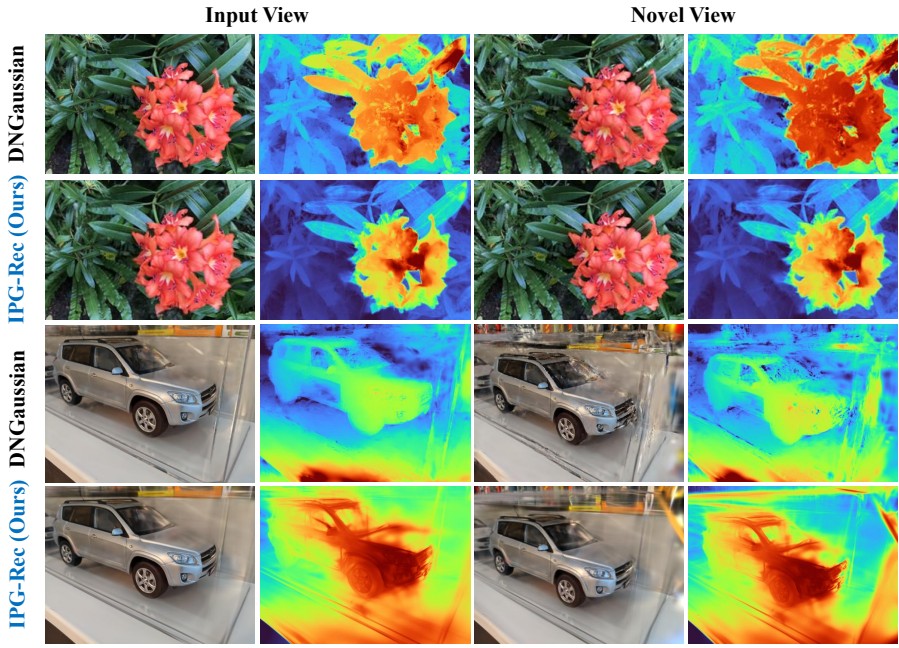

Figure 7: **Comparison of Geometric Regularization Approaches.** DNGaussian (Li et al., 2024a) utilizes depth map patches to regularize the Gaussian field. In contrast, our method uses object-level and global geometry supervision, resulting in cleaner edges and more coherence between input views and synthesized novel views.

