# OpenReview forum: "IPG-Rec: Instance-aware Progressive Geometry Rectification for High-fidelity Sparse Novel View Synthesis"
_ICLR.cc/2026/Conference — ICLR 2026 Conference Withdrawn Submission_

### Official Review · Reviewer_DZBP · 2025-10-28

**Soundness:** 1
**Presentation:** 3
**Contribution:** 2
**Rating:** 2
**Confidence:** 5

**Summary:**

This paper proposes IPG-Rec, a novel framework for sparse-view NVS based on 3D Gaussian Splatting. The method introduces two main contributions: 1) Instance-aware progressive geometry rectification, using segmentation masks from SAM and depth priors from Depth Anything V2 to enforce local and global geometric consistency. 2) Progressive pseudo-view generation, where a one-step diffusion model synthesizes new viewpoints that are gradually added into training for geometry refinement. The approach aims to improve geometric fidelity under extremely sparse input views. Experiments on LLFF, IBRNet, and TnT show improved performance over state-of-the-art 3DGS-based baselines.

**Strengths:**

- Comprehensive experiments on multiple benchmarks with ablations.

- The paper is easy to follow.

**Weaknesses:**

- The paper’s most contributing diffusion-based pseudo-view strategy is the same in concept and pipeline to Difix3D+. It merely adopts Difix3D+ as a plug-in within a 3DGS pipeline. The remaining contributions of instance-level depth regularization are an incremental variant of DNGaussian’s patch-level depth regularization, using SAM masks instead of patches. Overall, the paper feels more like an engineering combination of existing ideas than an original new research contribution.

- The method heavily depends on SAM to generate instance masks. However, SAM’s segmentation granularity varies greatly with prompts and scene context. The paper fails to clarify how prompts are generated or how segmentation consistency across scenes is enforced. Without manual correction or tuning, this component introduces non-determinism and potential manual bias, which breaks reproducibility and weakens its significance and fairness because human adjustments are involved.

- The instance-level depth regularization implicitly assumes clear, separable object instances. For cluttered, texture-dominant, or background-heavy scenes, SAM tends to over-segment, resulting in noisy or fragmented masks. The paper does not show quantitative or qualitative evidence that the method performs effectively and robustly in such settings.

- Due to *data leakage*, the reported results cannot be considered a fair comparison to prior work. The paper explicitly uses *validation views* to generate pseudo-views that are then added to the training set (Sec. 3.3 and Algorithm 1). This constitutes a serious form of data leakage between validation and training splits. The authors simply transfer this paradigm from Difix3D+, but were not aware that it's a huge soundness problem. These validation views should be strictly kept invisible during training. It means the model directly enhanced the target viewpoints during 3DGS optimization, artificially inflating evaluation scores.

- From the ablation study of Table 3, it's shown that the most contributing component is the progressive pseudo views, which is actually the simple employment of Difix3D+ and has data leakage problem. Except for it, the performance is not much beyond the baselines, even the DNGaussian and FSGS with the much worse MiDas depth prior.

**Questions:**

See the weaknesses.

---

### Official Review · Reviewer_MYDR · 2025-10-31

**Soundness:** 2
**Presentation:** 3
**Contribution:** 2
**Rating:** 2
**Confidence:** 5

**Summary:**

This work introduces geometry-based regularization to optimize 3D Gaussians, incorporating both scene-level and instance-level depth constraints to achieve more accurate appearance rendering. In addition, a diffusion model is used to generate pseudo labels for sampled views, providing supplementary information to enhance the 3D scene representation. Experimental results demonstrate that the proposed method outperforms existing approaches.

**Strengths:**

1) This work is well-organized and clearly written.
2) The proposed instance-level depth regularization is novel, encouraging pixels belonging to the same object to be spatially coherent.
3) The proposed method outperforms state-of-the-art methods.

**Weaknesses:**

1) The overall novelty of the work appears limited: (1) the proposed scene-level depth regularization closely resembles the approach introduced in SparseGS [A]; (2) the diffusion-based progressive pseudo-view geometry refinement largely follows the algorithm presented in DIFIX3D+ [B].
2) The paper claims to enable cross-view consistency, but it is unclear how this is achieved. The method relies only on monocular relative depth regularization, and SAM predicts masks per image without enforcing multi-view consistency.
3) While the work introduces instance-level depth regularization to enforce depth consistency within an object, Fig. 7 shows inconsistent depth estimates on the car, which raises questions about the effectiveness of this component.

[A] Xiong, Haolin, et al. "Sparsegs: Sparse view synthesis using 3d gaussian splatting." 3DV 2025.

[B] Wu, Jay Zhangjie, et al. "Difix3d+: Improving 3d reconstructions with single-step diffusion models." CVPR 2025.

**Questions:**

Please see the weaknesses.

---

### Official Review · Reviewer_znBe · 2025-11-01

**Soundness:** 3
**Presentation:** 3
**Contribution:** 2
**Rating:** 4
**Confidence:** 5

**Summary:**

This paper focuses on novel view synthesis using 3D Gaussian Splatting (3DGS) under sparse-view settings. The core idea is to leverage instance-level information to constrain scale alignment between monocular prior depth and 3DGS depth. Based on this, the method introduces pseudo-view supervision. Experiments are conducted on LLFF, TNT, and IBRNet datasets.

**Strengths:**

1. The proposed instance-level constraints are intuitively reasonable, as they effectively align monocular prior depth with 3DGS depth at the object level, which helps improve geometric consistency in sparse-view scenarios.
2. The paper is clearly written and well-structured, making it easy to follow the methodology and experimental setup.
3. Experiments demonstrate that the method achieves strong quantitative and qualitative performance compared to existing approaches.

**Weaknesses:**

1. The ablation studies are not fully convincing. The contribution of the instance-level constraints should be directly compared to patch-level constraints to clearly demonstrate their effectiveness. In the current ablations, the improvement from instance-level constraints appears very limited, with most gains coming from the use of pseudo-view supervision and the diffusion model.
2. In Appendix Tables 5 and 6, the proposed method performs poorly on the leaves scene, which contains many fine-grained structures. It is unclear whether the method’s design negatively affects the preservation of fine details, for example, due to the lack of a robust mechanism to handle noise from the diffusion model outputs. Further clarification on this point is needed.

**Questions:**

I would like the authors to clarify the actual performance gain provided by the instance-level constraints, and to discuss the robustness of the method to noise in the diffusion model outputs.

---

### Note · Authors · 2025-11-14

**Comment:**

I have confirmed the withdrawal.

**Withdrawal Confirmation:**

I have read and agree with the venue's withdrawal policy on behalf of myself and my co-authors.